# Narrative Review of Chronic Inflammation in Uterine Myoma: Lack of Specialized Pro-Resolving Lipid Mediators (SPMs) and Vitamin D as a Potential Reason for the Development of Uterine Fibroids

**DOI:** 10.3390/biomedicines13081832

**Published:** 2025-07-26

**Authors:** Pedro-Antonio Regidor, Manuela Mayr, Fernando Gonzalez Santos, Beatriz Lazcoz Calvo, Rocio Gutierrez, Jose Miguel Rizo

**Affiliations:** 1Exeltis Pharmaceuticals Holding S.L., Adalperostr. 84, 85737 Ismaning, Germany; 2Exeltis Germany, Adalperostr. 84, 85737 Ismaning, Germany; manuela.mayr@exeltis.com; 3Independent Researcher, Concepcion Arenal n2, 13600 Alcázar de San Juán, Spain; fgs9mail@gmail.com; 4Exeltis Pharmaceuticals Holding S.L., Manuel Pombo Angulo 28, 28050 Madrid, Spain; beatriz.lazcoz@exeltis.com; 5OTC Chemo, Manuel Pombo Angulo 28, 28050 Madrid, Spain; rocio.gutierrez@chemogroup.com (R.G.); josemiguel.rizo@chemogroup.com (J.M.R.)

**Keywords:** uterine myoma, inflammation, vitamin D, specialized pro-resolving mediators (SPMs)

## Abstract

Uterine leiomyoma (uterine fibroids, UF) are benign myometrium tumors that affect up to 70% of the female population and may lead to severe clinical symptoms. Despite the high prevalence, pathogenesis of UF is not understood and involves cytokines, steroid hormones, and growth factors. Additionally, an increased deposition and remodelling of the extracellular matrix is characteristic for UF. Vitamin D seems to play a new role in UF. Interestingly, hypovitaminosis D correlates with a higher prevalence of myomas and the severity of the myomas. Administration of vitamin D in women with insufficiency (serum level <30 ng/mL) restored the vitamin D status and reduced the mild symptoms of myomas. In addition, inflammatory processes may play a role. In the past years, it has become clear that cessation of inflammation is an active process driven by a class of lipid mediator molecules called specialized pro-resolving mediators (SPM). Inadequate resolution of inflammation is related to several chronic inflammatory diseases and several studies have proven the crucial role of SPMs in improving these diseases. In this review, we will give an overview on processes involved in UF growth and will give an overview on the modern view regarding the concept of inflammation and the role of SPMs in resolution of inflammation, especially in chronic inflammatory diseases.

## 1. Introduction

Uterine leiomyoma (myoma or uterine fibroids [UF]) are the most common nonmalignant pelvic tumors during the reproductive span [1] and up to 70% of women are affected. In most cases, myoma do not cause symptoms and grow very slowly [2]. However, about 25% of the affected patients suffer from clinically relevant symptoms [2,3]. The incidence of UF is especially high among African-American women [4,5], who also suffer from more severe symptoms than the Caucasian population [6]. Typical symptoms include heavy uterine bleeding, dysmenorrhea, dyspareunia, and constipation. In addition, uterine leiomyoma can also be a cause of infertility or abortion [2,7].

Myoma were described as monoclonal tumors of the smooth musculature. However, recent research on the Mediator Complex Subunit 12 gene, *MED12(+)* and *MED12(−)* fibroid variants, presents them, by single cell RNA sequencing, as rather heterogeneous cell types [8]. The tumors, although benign, are characterized by abnormal vascularization. Initiation and progression is driven by a variety of factors including local hypoxia and gonadal steroids as well as locally expressed angiogenic growth factors including EGF (epidermal growth factor) and VEGF (vascular endothelial growth factor). Cytokines and chemokines that are also involved in the pathogenesis not only promote the proliferation of cells but also the deposition of ECM (extracellular matrix) [9], which is the main component of UF and an important factor in its development [1,9]. There also seems to be a link between myoma formation and genetic aberrations, as cytogenetic tests showed abnormalities in 40% of the samples tested: 65% overexpressed the *HMGA2* (high-mobility group AT-hook 2) gene. Furthermore, 70% of leiomyoma samples carried mutations in the subunit of a transcription regulator [10,11].

Estrogens and progesterone (P4) play a role in the initiation and propagation of UF. Progesterone receptor A and B (PR-A and PR-B) expression is induced by estrogen signaling at the estrogen receptor (ER). This, in turn, increases the effectiveness of progesterone. As a result, the progesterone-induced increase in mitotic activity of UF cells increases [12,13]. Additional effects of growth factor regulation or microRNA expression and ECM accumulation seem to be mediated by P4 [1,13,14,15]. Progesterone is known to induce cell proliferation in UF tissue, while reducing apoptosis. Its activity is mediated by growth factors such as TGFβ (transforming growth factor β), although the action of these mediators would not occur in the absence of P4 [16]. In addition, the proliferation and growth of myoma may be induced and promoted by local and systemic inflammatory processes. Although the question is controversially debated [10,17,18], it is assumed that chronic inflammation promotes the pathogenesis of leimyoma in the following review.

Tumor types of various etiology, including uterine fibroids, are associated with an inflammatory microenvironment, which is induced by pro-inflammatory cytokines produced by immune cells and undifferentiated cells [17]. Interestingly, pro-inflammatory cytokines (such as interleukin 6 and 1 (IL-6 and IL-1) and tumor necrosis factor alpha) were shown to increase estrogen synthesis via modulation of aromatase, estradiol 17-β-dehydrogenase, and estrone sulfatase activity. Inhibition of the synthesis of prostaglandin E2 was also demonstrated to exert an influence on this balance [19,20]. As mentioned above, ECM remodeling and accumulation is characteristic of inflammatory processes and is a key factor in the development of UF. It is associated with excessive collagen accumulation and a reduced number of micro vessels, which in turn promote degenerative atrophic changes of the myometrium with cellular senescence and involution. The collagenized areas exhibit a hyaline and hypocellular morphology [21]. ECM generation is driven by cytokine- and growth factor- signaling; on the other hand, it acts as a source for the continuous availability of these molecules as it exerts a stabilizing effect.

Immune cells, especially monocytes and macrophages, responding to these inflammatory signals stimulate fibrotic changes of the tumor tissue. The TGF-β is one of the major players in these processes [22,23], but also TNF-α and the granulocyte-macrophage colony-stimulating factor, as well as interleukins (IL) 1, IL6, IL11, IL13, IL15, and IL13, were found to be involved in the pathogenesis of UF. Drugs that are commonly used in treatment of UF may interfere with these inflammatory processes and it may by worthwile to gain more understanding about these interactions [7].

Another aspect in the development of uterine myoma is the role of oxydative stress. The imbalance between prooxidants and antioxidants may induce the growth of uterine myoma. This imbalance acts via the interconnected cascades that involve angiogenesis, hypoxia, dietary factors, and smoking/environmental exposures, among others.

Oxidative stress in turn drives fibroid development through its impact on genetics, epigenetics, and fibrogenesis [24].

Reactive oxygen species (ROS) have been associated with the abovementioned promotion of the *MED12* mutations in the development of these tumors in vitro [25]. This phenomenom can be added to the pathophysiological mechanisms of uterine fibroids.

The purpuse of this review is to analyze the possible interactions between the use of vitamin D and the mono and dihydroxylates of omega-3 fatty acids in their modulating role in inflammation. This narrative review was conducted using open access literature.

## 2. Role of Vitamin D

Lima et al. recently showed [26] a reduced vitamin D receptor (VDR) expression in leiomyoma tissue compared with myometrial tissue, which can be associated with the pathogenesis and development of human uterine leiomyomatosis. The cause of the reduction in the expression of these receptors is still unknown and it is also not clear whether it is an event that occurs in parallel to the onset or progression of uterine leiomyomas. Furthermore Halder et al. [27] analyzed the influence of vitamin D when studying the effect of 1,25-dihydroxyvitamin D3 on fibrosis-related protein expression in TGF-3 induced uterine leiomyoma cells in vitro. Myoma cells were treated with TGFß-3 with or without vitamin D. They identified that TGF-3 induced the expression of fibronectin and collagen protein type 1 in myoma cells, which was suppressed by vitamin D and considered as an antifibrotic factor in the treatment of benign uterine myomas. Halder et al., in a subsequent study [28], investigated the risk of benign uterine tumors in regard to VDR protein and determined the biological function of 1.25(OH)_2_D_3_ in the regulation of proteins related to the extracellular matrix, which is essential in the formation of leiomyomas. They identified reduced VDR levels in more than 60% of the uterine tumors analyzed compared to the adjacent myometrium. In fact, the levels of VDR in the uterine myoma samples were significantly lower than the levels in the adjacent myometrial samples. Al-Hendy et al. [29], who investigated the role of 11.25(OH)_2_D_3_ in the expression of sex steroid receptors in leiomyoma cells, realized that the deregulation of steroid hormones and their receptors could be a starting point for myoma growth since 1.25(OH)_2_D_3_ VDR expression acts as an antiesterogenic agent in these cells. They also showed a significant decrease in estrogenic receptor levels in leiomyoma cells treated with 1.25(OH)_2_D_3_ and analyzed for receptor expression and location. In contrast, 1.25(OH)_2_D_3_ induced the expression of its own VDR, suggesting that 1.25(OH)_2_D_3_ acts as an antagonist of hormone receptors with antiestrogenic and antiprogesteronic functions. Paffoni et al. [30] analyzed serum levels of vitamin D in women with myoma and recognized that the vitamin D concentration was significantly lower in women with myomas compared to women in the control group (11.1 and 18.0 ng/mL, respectively; *p* < 0.010 and OR = 2.2). Similar results were obtained by Baird et al. [31], who assessed vitamin D and the risk of uterine myomas and found that women with sufficient vitamin D had an estimated 32% reduction in the incidence of myomas com- pared with those with insufficient vitamin D. Sabry et al. [32] studied whether the low serum levels of vitamin D were associated with the increased risk and occurrence of uterine myomas and found that reduced serum levels of 1.25(OH)_2_D_3_ vitamin D were significantly associated with the occurrence of myomas. A statistically significant inverse correlation was also observed between the serum levels of 25-(OH) vitamin D and the total leiomyoma volume within the case cohort. The above-described studies indicate that the loss of vitamin D functions due to the reduction of vitamin D3 levels and/or reduced expression of the VDR may be associated with the growth and development of different types of neoplastic lesions. It reinforces the hypothesis that low *VDR* expression may be associated with the growth and development of myomas, presenting itself as an important biomarker in this pathology. Vitamin D is believed to regulate cell proliferation and differentiation, reduce angiogenesis, and stimulate apoptosis.

Nowadays, hypovitaminosis D is supposed to be a major risk factor in the development of fibroids. In many studies vitamin D appears to be a powerful factor against UFs, resulting in inhibition of tumor cell division and a significant reduction in its size; however, the exact role of this compound and its receptor in the pathophysiology of myomas is not fully elucidated. Based on available studies, vitamin D and its analogs seem to be promising, effective, and low-cost compounds in the management of myomas and their clinical symptoms and the anti-tumor activities of vitamin D play an important role in uterine myoma biology.

Single nucleotide polymorphisms (SNPs) and vitamin D modulation have been discribed. An association has been established between uterine fibroids and the variants rs739837 and rs886441 in the nuclear hormone receptor for vitamin D [33]. Shahbazi et al.’s research supports this idea that the *VDR* rs2228570 variant is related to uterine fibroids; specifically, there is a correlation between the *VDR* TT genotype and an elevated risk of uterine fibroid occurrence [34]. Further, Yilmaz et al. demonstrated that the presence of the rs2228570 CC genotype might act as a risk-reducing factor, while the T allele could potentially contribute to the risk of UL, aligning with Shahbazi’s findings [35].

However, one of the findings indicated that there is no observed correlation between the *VDR* variants rs731236, rs1544410, and rs2228570, and the incidence of uterine leiomyoma in Caucasian women, reinforcing the concept that SNPs linked to vitamin D metabolism and skin color are connected to the presence of uterine fibroids in African-American women [36].

Notably, among the scrutinized SNPs, rs12800438 near the DHCR7 gene and rs6058017 in the ASIP gene are implicated in vitamin D synthesis in the skin in African-American women [37].

These changes may also be due to the genetic aspects of uterine fibroids. Approximately 40% of UL have non-random and tumor-specific chromosome abnormalities. This has allowed classification of some UF into well-defined subgroups which include deletion of portions of 7q, trisomy 12, or rearrangements of 12q15, 6p21, or 10q22. Additional abnormalities, which appear consistently but not as frequently, include rearrangements of chromosomes X, 1, 3, and 13. The variety of chromosomal rearrangements, including but not limited to translocation, deletion, and trisomy, predict different molecular genetic mechanisms for UF formation and growth [38].

## 3. Inflammatory Processes and the Role of Lipid Acid-Derived Mediators

Acute inflammation occurs in response to a microbial infection or injury. The initiated processes are necessary to render pathogens innocuous, remove cell debris and restore affected tissue. Eicosanoid lipid mediator (LM) molecules that are synthesized from the ω-6 poly-unsaturated fatty acid (PUFA) arachidonic acid (AA) are characteristic pro-inflammatory signaling molecules. These prostanoids comprise prostaglandins (PG), leukotrienes (LT), and thromboxanes (TX), and are synthesized via the enzymes cyclooxygenase 1 and 2 (Cox-1/2) by cells of the innate immune system (granulocytes and macrophages), which are quickly recruited to a site of injury or infection [39]. Further pro-inflammatory cytokines including TNFα, IL-1, and Il-6 are secreted by mast cells, and classically activated M1 macrophages contribute to the inflammatory response and its characteristic symptoms (heat, swelling, redness, and pain). Prostanoids and cytokines also attract neutrophils and monocytes, which migrate towards the site of the event and enter the affected tissue, thereby further promoting the inflammatory response of the organism [40,41] Rapid and efficient initiation of inflammation is vital for the organism, but timely cessation of the processes is of equal importance, as excessive inflammation can lead to phenomena such as cytokine storm or sepsis, which are life-threatening events [42]. Chronic inflammatory pathologies like cardiovascular disease (CVD), diabetes, or autoimmune disorders are further examples of inadequate cessation of inflammation [43,44]. Resolution of inflammation is an active process that is driven by so-called specialized pro-resolving lipid mediator molecules (SPM), which are able to initiate and drive resolution in animal models [45,46]. Based on their chemical structure and biosynthetic pathways, they are clustered into four families: the resolvins (Rvs), protectins (PDs), lipoxins (LXs), and maresins (Mars) [45,47]. Rvs, PDs, and LXs derive from the ω-3 PUFAs EPA (eicosa pentaenoic acid) and DHA (docosahexaenoic acid) and their biosynthesis involves certain lipoxygenases, but also the COX enzymes. They are generated via the hydroxylated precursors 18HpETE, 17-HpDHA, and 14-HpDHA. ω-6 PUFA AA is the precursor for LX [47,48]. In Figure 1, an overview on the biosynthetic pathways is given. Notably, aspirin, which inhibits the COX enzymes irreversibly, blocks the synthesis of prostanoids by modifying the catalytic domains of COX. However, their capacity to catalyse the synthesis of the SPM precursors 18HpETE, 17-HpDHA, and 14-HpDHA is not abolished. The emerging SPMs are referred to as aspirin-triggered SPMs (AT-SPMs) and are not only used in experiments but, e.g., in modern nutraceuticals [46,47,48,49,50,51].

Eicosanoid lipid mediator (LM) molecules (LT, TX, PG, etc.) from arachidonic acid AA and pro-inflammatory cytokines induce the influx of PMNs and moncytes and lead to the classical signs of acute inflammation (calor, rubor, tumor, and dolor). At the very beginning, eicosanoids initiate the LM class switch resulting in the production of SPMs from omega-3 PUFAs (see Figure 1 and text). SPMs are critical for the resolution of the inflammation. LM: lipid mediator; LT: leukotriene, TX: thromboxane, PG: prostaglandins; PUFA: poly-unsaturated fatty acid, and SPM: specialized pro-resolving mediators.

## 4. SPMs Are Essential for Resolution of Inflammation

SPMs are crucial for the resolution of inflammation. How the resolution of inflammation is achieved is of the utmost importance. There are several things that must be achieved: recruitment of neutrophils must stop, dead cells must be eliminated, and macrophages need to switch towards the resolving M2 macrophages [52]. These and other key events in obtaining resolution are driven by SPMs, as reviewed in detail elsewhere [48,49,50,51,52,53,54]. In short, SPMs are necessary for stopping the recruitment of neutrophils and triggering apoptosis. Further, they play a role in the downregulation of pro-inflammatory cytokines, such as TNF-α, IL-6, IL-8, IL-12, and PAF (platelet-activating factor) and prostaglandin production. In fact, SPMs are involved in clearing the infection site and in tissue regeneration, since they promote efferocytosis and wound healing. In the process, these resolutive events are initiated at the same time as the inflammatory process starts, which is reflected by the interlinkage of prostaglandin synthesis with SPM biosynthesis. PGE2 and PGD2 are required to induce type 1 lipoxygenases, the key enzymes in the SPM biosynthetic pathways which produce LXs, Rvs, and PD2 [55]. Consequently, when PG synthesis is inhibited, the lipid mediator class switch from pro-inflammatory to pro-resolutive is interfered with, which may lead to delayed resolution [56,57]. To sum up, the inflammation always starts with its active resolution, since alpha signals omega in the whole cascade of signaling pathways [55]. In this way, a new perception of inflammation is shown in Figure 2.

Extensive research into SPMs led to detailed knowledge concerning the structure, biosynthetic pathways, and function of SPMs and different options for the outcome of inflammatory responses are dealt with in Serhan et al., Bandeira-Melo et al. and Levy et al. [45,48,56,57]. Figure 3 shows the current view. Should the resolution of inflammation not take place adequately and remain “switched on”, a continuous production of pro-inflammatory signaling molecules and thus a state of chronic inflammation may be established. In addition to their activity in promoting the resolution of inflammation, the SPMs represent a link to the adaptive immune response. For example, lipoxin LX3 also activates natural killer cells (NKC) [58]. The differentiation of CD4+ T cells was shown to also be modulated by resolvins RvD1 and RvD2 and maresin Mar1 [59].

## 5. The Significance of SPMs in Chronic Inflammatory Diseases

Animal experiments have demonstrated the role of SPMs in the prevention or improvement of chronic inflammatory diseases. In the case of periodontitis, pathogenesis is based on an inflammatory process resulting from an infection with P. gingivalis that can be mimicked in air pouche mouse models. In these experiments, increased recruitment of neutrophils and upregulation of COX-2 enzymes was observed, which was decreased by supplementation with lipoxin LXA analogues. [60]. A further model in mice under the influence of 12/15 lipoxygenase demonstrated the role of the SPMs lipoxygenase A (LXA) and Resolvin D (RvD) in the pathogenesis of atherosclerosis, which is considered a chronic inflammatory disease [61]. In mice under the influence of 12/15 lipoxygenase, the increased production of RvD, PD, and 17-HpDHA was protective, as the development of atherosclerosis was reduced in comparison to wild type mice. The anti-atherogenic activity of LXA, PD, and RvD derived from different processes such as reduced expression of endothelial adhesion molecules and diminished secretion of cytokines. Notably, this mouse model also demonstrated the role of nutrition in the pathogenesis of atherosclerosis, since a standard high-fat western diet rendered the transgenic mice as susceptible to atherosclerosis as the wild type animals [61,62]. In a rat model for arthritis, a further common chronic inflammatory disease, resolvin RvD, as well as its precursor metabolite (17-HDHA) led to a greater reduction in pain and tissue damage compared with medication with steroids [63]. Fibrosis, which is also a characteristic feature of uterine leiomyoma, may also be considered a consequence of an inappropriate resolution of inflammation. For pulmonary fibrosis, the experimental LX4 analogues (AT-LX4) that were administered exogenously reduced fibrosis in animal models [64]. LXA4 and benzo-LXA4, an analogue, also reduced the extent of fibrotic changes in kidneys in a rat model of early renal fibrosis [65], and for the resolvin RvE1, antifibrotic effects were demonstrated in a mouse model of obstructed kidney [66]. Although the role of SPMs in UF development has not been elucidated yet, there are some shared features with other chronic inflammatory diseases in which the role of in-adequate resolution has been demonstrated. Further research in this area may lead to new treatment options.

The limitations of the reviewed research data are the lack of clinical formulations containing vitamin D and the derivates of the omega-3 fatty acids used in the management of uterine fibroids.

The first attempts are made to close this gap by introducing such compounds into the market.

Therefore, further research on the role of inflammation in general and SPMs might yield further important knowledge on uterine fibroids.

## 6. Conclusions

The pathogenesis of uterine fibroids has not been fully clarified yet. Cytokine, growth factor, and steroid hormone signaling together with inflammatory processes may be involved. Vitamin D acts on human reproduction not only by calcium homeostasis, but also due to its paramount importance as a direct regulator of aromatase gene expression. Accumulating evidence suggests that VDR-bound 1.25(OH)_2_D_3_ acts as a transcription factor to regulate the expression of the *CYP19* gene. This gene encodes the fat and ovarian tissue aromatase. Hence, this mechanism encodes estrogen regulation. Vitamin D is also able to stimulate the production of progesterone, different estrogens, and substances like the insulin-like growth factor binding protein 1 (IGFBP-1), especially in cultured human ovarian cells. Altogether, these different modes of vitamin D influence hormone-dependent benign tissues like, e.g., uterine fibroids, as they have a modulating effect on growth and differentiation [67]. Other estrogen-dependent diseases like endometriosis seem to also be modulated by vitamin D. Activated CD4 and CD8 cells widely express vitamin D receptors and both the activated and metabolized enzymes, 1-a-hydroxylase and 24-hydroxylase. This induces local vitamin D expression and has a negative effect on the potential regression of endometriosis implants. In addition, in a large prospective cohort study, a high plasma 25(OH)D level was associated with a lower risk of endometriosis [68]. Finally, vitamin D showed inhibitory capacities on the expression of ER and PR in a dose-dependent manner in leiomyoma cells. [69]. As has been described during the last years, active resolution of inflammation, which is mainly driven by specialized pro-resolving lipid mediator molecules (SPMs), is crucial for the adequate cessation of acute and chronic inflammation, as demonstrated for diseases like periodontitis, atherosclerosis, or diabetes.

A second positive mechanism in the development of uterine fibroma can be the use of dietary antioxidants and 3-hydroxy-3-methyl-gutaryl-coenzyme A (HMG-CoA), as they can influence the development of oxidate stress [25].

The role of SMPs and their influence on uterine myoma growth still needs to be elucidated. Nevertheless, in a subset of women with erythrocyte FA measurements, a lower odds of fibroids among women with higher n-3 PUFA erythrocyte levels and a greater odds among those with higher trans FA erythrocyte levels could be observed, suggesting n-3 PUFAs and trans FAs may be associated with fibroids [70]. In a preliminary study (unpublished data) a reduction in the bleeding-associated pain could also be observed.

## Figures and Tables

**Figure 1 biomedicines-13-01832-f001:**
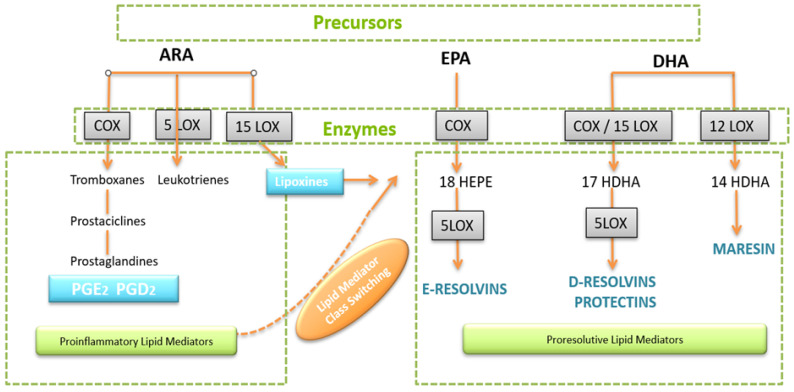
Biosynthesis of the SPMs, resolvins, protectins, and maresins. EPA, eicosapentaenoic acid; 18-HpEDE, 17-HpDHA, and 14-HPDHA: precursors of the SPMs during biosynthesis; Cox-1/2: cyclooxygenases; and LOX: lipoxygenase Aspirin triggers biosynthesis of 18-HpEDE and 17-HpDHA intermediates via modification of COX enzymes. Maresins are produced by macrophages via a preliminary lipoxygenation step. Further lipoxygenases are required for SPM biosynthesis as depicted (modified from Serhan [48]).

**Figure 2 biomedicines-13-01832-f002:**
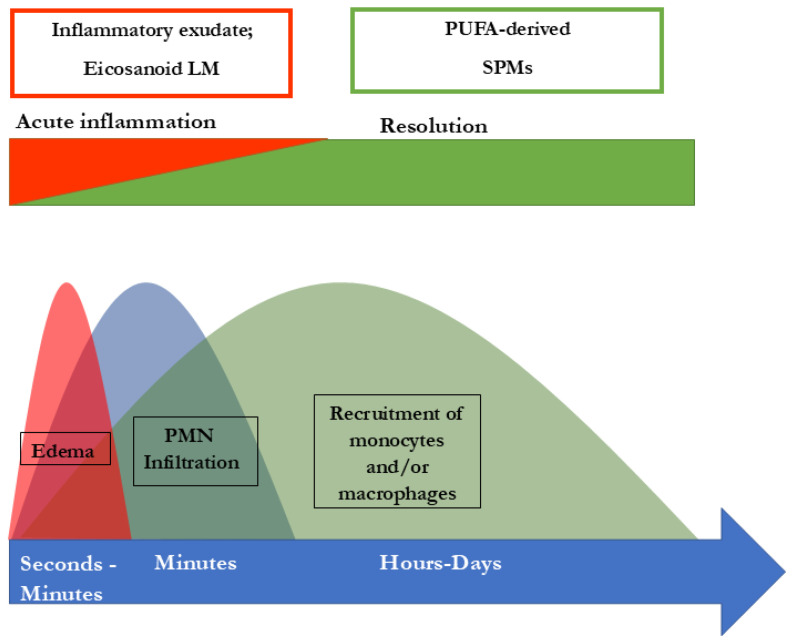
Concomitant initiation and resolution of inflammation. Synthesis of pro-resolving lipid mediator molecules is initiated at the beginning of inflammatory processes. LM: lipid mediator; PUFA: poly-unsaturated fatty acid, PMN: polymorphonuclear neutrophils; and SPM: specialized pro-resolving mediators (maresins, resolvins, protectins, and lipoxins). Modified from Serhan and Levy [54].

**Figure 3 biomedicines-13-01832-f003:**
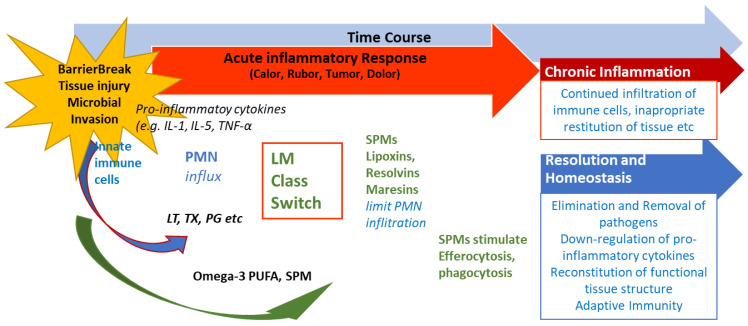
Role of SPMs during resolution of inflammation and potential outcome of inflammatory processes. After inflammation is triggered, cells of the innate immune system synthesize eicosanoid lipid mediator (LM) molecules (LT, TX, PG, etc.) from AA and pro-inflammatory cytokines that stimulate the influx of PMNs and monocytes and result in the classical signs of acute inflammation (calor, rubor, tumor, and dolor). At the beginning of inflammation, eicosanoids trigger the lipid mediator (LM) class switch resulting in the synthesis of SPMs from omega-3 PUFAs (see Figure 1 and text). SPMs are crucial for resolution of inflammation. LM: lipid mediator; LT: leukotriene, TX: thromboxane, PG: prostaglandins; PUFA: poly-unsaturated fatty acid, and SPM: specialized pro-resolving mediators (modified from Serhan [48,50]).

## Data Availability

No new data were created or analyzed in this study.

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
