# Peer review of "Narrative Review of Chronic Inflammation in Uterine Myoma: Lack of Specialized Pro-Resolving Lipid Mediators (SPMs) and Vitamin D as a Potential Reason for the Development of Uterine Fibroids"

_biomedicines, 2025, doi:10.3390/biomedicines13081832_

Round 1

Reviewer 1 Report

Comments and Suggestions for Authors

Congratulations to the authors! This article presents a well structured exploration of the roles of chronic inflammation, SPMs, and vitamin D in the development of uterine fibroids. It offers a promising perspective by suggesting that inadequate resolution of inflammation may contribute to the tumor pathogenesis, a hypothesis not widely explored so far.

The manuscript would benefit from clearer language and more concise phrasing, as some sentences are overly complex or inconsistent. Terms such as "gen" instead of "gene" and "ostrogen" instead of "estrogen" should be corrected. Also, while the discussion of SPMs is informative, the link between these mediators and uterine fibroids remains unclear and could be strengthened by referencing disease-specific evidence.

The conclusion can be more impactful if it is more concise and better focused on the key words, vitamin D’s dual regulatory role in hormone balance and the therapeutic promise of SPMs.

Overall, this article represents a meaningful contribution to the field, and with minor revisions to the language and clearer connections between SPMs and fibroid biology, it would be suitable for publication.

Author Response

Point to point replay

Dear reviewers:

Thank you, a lot, for the absolutely very valuable and constructive comments.

I think that I will be able to introduce all the required changes

In the red the answers:

Reviewer 1

Congratulations to the authors! This article presents a well structured exploration of the roles of chronic inflammation, SPMs, and vitamin D in the development of uterine fibroids. It offers a promising perspective by suggesting that inadequate resolution of inflammation may contribute to the tumor pathogenesis, a hypothesis not widely explored so far.

The manuscript would benefit from clearer language and more concise phrasing, as some sentences are overly complex or inconsistent. Terms such as "gen" instead of "gene" and "ostrogen" instead of "estrogen" should be corrected. Also, while the discussion of SPMs is informative, the link between these mediators and uterine fibroids remains unclear and could be strengthened by referencing disease-specific evidence.

I have changed accordingly

The conclusion can be more impactful if it is more concise and better focused on the key words, vitamin D’s dual regulatory role in hormone balance and the therapeutic promise of SPMs.

I have done this

Overall, this article represents a meaningful contribution to the field, and with minor revisions to the language and clearer connections between SPMs and fibroid biology, it would be suitable for publication.

Reviewer 2 Report

Comments and Suggestions for Authors

The work is devoted to the consideration of topical issues of the pathogenesis of uterine fibroids and, in particular, examines modern literature data on the processes involved in the growth of uterine fibroids, examines the modern view of the concept of inflammation and the role of SPMS in the resolution of inflammation, especially in chronic inflammatory diseases. There are the following suggestions for this work: 1. The genes should be italicized (for example, line 49). 2. The introduction is presented in one large paragraph of 1.5 pages. The introduction should be divided into several paragraphs according to the generality of the material considered in them. 3. The introduction should end with an indication of the purpose of this work. The purpose of the work should be clearly defined. 4. Are there any features of vitamin D metabolism in African Americans that would explain the high incidence of uterine fibroids in this ethnic group? These data should be given in the "Role of Vitamin D" section. 5. In each of the sections presented in the work (2, 3, 4 and 5), genetic data (GWAS data and data from associative studies) should be provided, which would show the relationship of genetic determinants involved in vitamin D metabolism, inflammatory processes, etc., with the risk of uterine fibroids. These data would explain the "root causes" of the links shown by the authors in the work of vitamin D metabolism, inflammatory diseases, etc. with uterine fibroids. 6. Information about the limitations of this study should be added to the paper.

Author Response

Dear reviewer:

Thank you, a lot, for the absolutely very valuable and constructive comments.

I think that I will be able to introduce all the required changes

In the red the answers:

Reviewer 2

The work is devoted to the consideration of topical issues of the pathogenesis of uterine fibroids and, in particular, examines modern literature data on the processes involved in the growth of uterine fibroids, examines the modern view of the concept of inflammation and the role of SPMS in the resolution of inflammation, especially in chronic inflammatory diseases. There are the following suggestions for this work:

  1. The genes should be italicized (for example, line 49).

Done

  1. The introduction is presented in one large paragraph of 1.5 pages. The introduction should be divided into several paragraphs according to the generality of the material considered in them.

Done

  1. The introduction should end with an indication of the purpose of this work. The purpose of the work should be clearly defined.

Done and introduced in the new version of the manuscript

  1. Are there any features of vitamin D metabolism in African Americans that would explain the high incidence of uterine fibroids in this ethnic group? These data should be given in the "Role of Vitamin D" section.

Done and introduced in the new version of the manuscript

  1. In each of the sections presented in the work (2, 3, 4 and 5), genetic data (GWAS data and data from associative studies) should be provided, which would show the relationship of genetic determinants involved in vitamin D metabolism, inflammatory processes, etc., with the risk of uterine fibroids. These data would explain the "root causes" of the links shown by the authors in the work of vitamin D metabolism, inflammatory diseases, etc. with uterine fibroids.

Done and introduced in the new version of the manuscript

  1. Information about the limitations of this study should be added to the paper.

Done and introduced in the new version of the manuscript

Reviewer 3 Report

Comments and Suggestions for Authors

  • The article addresses the pathogenesis of a common disease in the population. It will be of value to researchers in the field.
  • The abstract is representative of the study. Keywords are appropriate.
  • The title needs to include the type of review. i.e, narrative, scoping, systematic, umbrella
  • The introduction adequately provides an overview of existing literature and identifies the gap.
  • It is suggested to follow a format for (academic) writing. The aims and objectives of the study need to be mentioned. After classifying the type of review, a brief section on methodology or the process of gathering evidence needs to be included. This will ensure reproducibility of the research and minimize bias.
  • The figures are appropriately used and are representative of the data. They are easy to understand. The quality of Figure 1 needs to be improved.
  • The conclusion section is appropriate, but will benefit from including strengths of the study and practical implications.
  • Minor modifications-
  • Line 45- strong uterine bleeding, dysmenorrheal- should be corrected to heavy menstrual bleeding and dysmenorrhea
  • Line 105- Remove Role of Vitamin D from the beginning of the sentence
  • Line 112-115- Is it TGF3 or TGF β3?
  • 25 (OH) should be corrected to 1,25 (OH)2
  • Line 126- What is meant by similar to the results of the present study?
  • Line 156-157- Add a reference
  • Line 195- protectins (PD) should be (PDs)
  • Line 213-216- LM: Lipid…. Should be included in the description in Figure 1.
  • Expand terms like MED, ECM, ER, IL, VDR in the text at the first mention.
  • The references are appropriate and well-cited. However, there are inconsistencies in the style, especially the use of et al after a certain number of author names. A significant number of references are more than 20 years old. It can be modified to a uniform style/ replaced according to the journal guidelines.

Author Response

Dear reviewer:

Thank you, a lot, for the absolutely very valuable and constructive comments.

I think that I will be able to introduce all the required changes

In the red the answers:

Reviewer 3

The article addresses the pathogenesis of a common disease in the population. It will be of value to researchers in the field.

The abstract is representative of the study. Keywords are appropriate.

The title needs to include the type of review. i.e, narrative, scoping, systematic, umbrella

The introduction adequately provides an overview of existing literature and identifies the gap.

It is suggested to follow a format for (academic) writing.

The aims and objectives of the study need to be mentioned.

I have introduced this at the end of introduction

After classifying the type of review, a brief section on methodology or the process of gathering evidence needs to be included. This will ensure reproducibility of the research and minimize bias.

Changed

The figures are appropriately used and are representative of the data. They are easy to understand.

The quality of Figure 1 needs to be improved.

I have introduced a new figure 1

The conclusion section is appropriate, but will benefit from including strengths of the study and practical implications.

Minor modifications-

Line 45- strong uterine bleeding, dysmenorrheal- should be corrected to heavy menstrual bleeding and dysmenorrhea

Line 105- Remove Role of Vitamin D from the beginning of the sentence

Line 112-115- Is it TGF3 or TGF β3?

25 (OH) should be corrected to 1,25 (OH)2

Line 126- What is meant by similar to the results of the present study?

Line 156-157- Add a reference

Line 195- protectins (PD) should be (PDs)

Line 213-216- LM: Lipid…. Should be included in the description in Figure 1.

Expand terms like MED, ECM, ER, IL, VDR in the text at the first mention.

Done for all the points

The references are appropriate and well-cited. However, there are inconsistencies in the style, especially the use of et al after a certain number of author names. A significant number of references are more than 20 years old. It can be modified to a uniform style/ replaced according to the journal guidelines.

Round 2

Reviewer 2 Report

Comments and Suggestions for Authors

The authors tried to make the necessary adjustments to the work. However, there are still a few minor requests for work.: 1. Italicize not the word "gene" (as the authors did, for example, in line 59 and in the conclusion), but the names of the genes, for example, MED 12(+) (line 48), HMGA2 (line 58), etc. 2. The introduction should not include the names of subsections (paragraphs). 3. The paragraph (lines 147-154) should be deleted or moved to another place, as it does not correspond in meaning to the material presented in this subsection - the role of vitamin D in uterine fibroids. 4. Information about the limitations of the study should be provided before the conclusion section (and not in the conclusion section)

Author Response

The responses are in the attachment.

The authors tried to make the necessary adjustments to the work. However, there are still a few minor requests for work.: 1. Italicize not the word "gene" (as the authors did, for example, in line 59 and in the conclusion), but the names of the genes, for example, MED 12(+) (line 48), HMGA2 (line 58), etc.
Done
2. The introduction should not include the names of subsections (paragraphs).
The names of the subsections have been deleted.
3. The paragraph (lines 147-154) should be deleted or moved to another place, as it does not correspond in meaning to the material presented in this subsection - the role of vitamin D in uterine fibroids.
This paragraph has been moved done in the text.
4. Information about the limitations of the study should be provided before the conclusion section (and not in the conclusion section)
The paragraph of the limitations has been moved up in the text before the chapter conclusions.
